# Impact of Respiratory Dust on Health: A Comparison Based on the Toxicity of PM2.5, Silica, and Nanosilica

**DOI:** 10.3390/ijms25147654

**Published:** 2024-07-12

**Authors:** Aoxiang Hu, Rou Li, Guo Chen, Shi Chen

**Affiliations:** Key Laboratory of Molecular Epidemiology of Hunan Province, Hunan Normal University, Changsha 410013, China; 202320193563@hunnu.edu.cn (A.H.); lirounines@hunnu.edu.cn (R.L.); chenguo777@hunnu.edu.cn (G.C.)

**Keywords:** PM2.5, silica, nanosilica, toxicity, health implications

## Abstract

Respiratory dust of different particle sizes in the environment causes diverse health effects when entering the human body and makes acute or chronic damage through multiple systems and organs. However, the precise toxic effects and potential mechanisms induced by dust of different particle sizes have not been systematically summarized. In this study, we described the sources and characteristics of three different particle sizes of dust: PM2.5 (<2.5 μm), silica (<5 μm), and nanosilica (<100 nm). Based on their respective characteristics, we further explored the main toxicity induced by silica, PM2.5, and nanosilica in vivo and in vitro. Furthermore, we evaluated the health implications of respiratory dust on the human body, and especially proposed potential synergistic effects, considering current studies. In summary, this review summarized the health hazards and toxic mechanisms associated with respiratory dust of different particle sizes. It could provide new insights for investigating the synergistic effects of co-exposure to respiratory dust of different particle sizes in mixed environments.

## 1. Introduction

Respiratory dust, with its tiny particle size, complex chemical composition, diverse sources, and ease of inhalation, causes diverse health effects when entering the human body and causes acute or chronic damage through multiple systems and organs [1]. Over the past few decades, a large number of studies have revealed the harmful effects of three different particle sizes of dust, including PM2.5 (<2.5 μm), silica (<5 μm), and nanosilica (<100 nm), on the respiratory system and overall health. As a common pollutant in industrial production, silica has become a growing concern with the development of modern industrial technology. It can cause adverse health effects among workers in numerous industries, including metal mining, metallurgy, the construction industry, tunneling, coal mining, the sandblasting of jeans, and the manufacture of artificial stone worktops [2]. Free silica with a diameter of less than 5 microns is able to penetrate deep into the bronchioles and alveoli of the lungs, leading to serious health problems such as silicosis in cases of long-term exposure [3]. PM2.5, defined as particles with aerodynamic diameters of less than 2.5 μm, is considered to be one of the major contributing factors to respiratory and cardiovascular diseases [4]. The tiny size confers it with the ability to penetrate deep into the cells of the lungs, thus triggering an inflammatory response and impairing lung function [5]. Meanwhile, with the rapid development of nanotechnology, nanosilica, as a novel material, is widely used in various fields, including medicine, cosmetics, food, and agriculture. The entire human system is more susceptible than ever to the effects of nanosilica-induced toxicity, resulting in epigenetic and phenotypic-related changes and damage to the immune system [6]. Due to the tiny size of nanosilica, it is believed to cause different negative complications compared to silica [6].

However, the specific toxic effects and mechanisms of PM2.5, silica, and nanosilica have not been systematically summarized. More in-depth studies are needed to understand the combined effects of simultaneous exposure to PM2.5, silica, and nanosilica in mixed environments and their corresponding mechanisms. In this study, we described the sources and characteristics of silica, PM2.5, and nanosilica. Based on their respective characteristics, we further explored the main toxicity induced by silica, PM2.5, and nanosilica in vivo and in vitro. Furthermore, we evaluated the health implications of respiratory dust on the human body, and especially proposed potential synergistic effects, considering current studies. In summary, this review summarized the health hazards and toxic mechanisms associated with respiratory dust of different particle sizes. It could provide new insights for investigating the synergistic effects of co-exposure to respiratory dust of different particle sizes in mixed environments.

## 2. Sources and Characteristics of PM2.5, Silica, and Nanosilica

PM2.5 is widely present in the atmosphere and is released from various sources, such as dust storms, forest fires, and volcanoes, as well as a variety of human activities, including transportation, fuel combustion, and industrial processes [7]. It is characterized by a small particle size and a large relative surface area, which makes it easy for it to adsorb various toxic substances [8]. Different climatic conditions, emission sources, and dispersion patterns make the sources and concentrations of PM2.5 vary significantly across locations [9]. In Europe, domestic biomass and fossil fuel combustion, including wood, coal, and natural gas for cooking and heating, are the major contributors to PM2.5. This leads to a significant increase in nitrate, sulfate, nitrogen oxides, and carbon oxides in the chemical composition of local PM2.5 [10]. In the United States, there are six major sources of particulate matter PM2.5, including biomass combustion, diesel vehicles, gasoline vehicles, dust, coal combustion, and metals. These sources are quite similar to the major contributors of PM2.5 in the European region [11]. In contrast, the history of PM2.5 monitoring in Asian countries is relatively short, and the main purpose of this monitoring is to understand the local air pollution situation. Therefore, it cannot be well used for epidemiological studies. As the largest and most populous country in Asia, China has experienced an overall deterioration in air quality over the past few decades to some extent. Therefore, analyzing the composition of PM2.5 in China could fill the gap in this area in Asia. A recent epidemiological study conducted in Xi’an showed that secondary constituents, combustion species, and transition metals were the most significant contributors to increased PM2.5 health risks, especially during the cold winter months [12].

Studies have shown that PM2.5 contains a wide range of chemical and biological substances that can enter the circulation and affect the entire organism [8,13]. Differences in the concentration and surface composition of PM2.5 from region to region can affect its capacity to trigger diseases. Therefore, Europe, the United States, and different countries in East Asia—including Japan, South Korea, and China—have established their own national air quality standards, respectively, which specify daily and annual average concentrations of PM2.5 (Table 1).

Among these regions, the United States and China further set up primary standards and secondary standards based on different objectives and application scenarios. The data come from the European Commission, the United States Environmental Protection Agency, the Ministry of the Environment Government of Japan, the Ministry of Environment of South Korea, and the Ministry of Ecology and Environment of the People’s Republic of China. However, due to the limitations of experimental conditions, most of the current experiments are built with PM2.5 models with almost no variation in surface composition. This makes it difficult for the experimental results to fully reflect the actual situation of local PM2.5 exposure. Future studies may be able to further correct the surface composition of PM2.5 to reduce the influence of this factor.

Silica is one of the most abundant minerals on Earth, comprising more than 60% of the Earth’s crust. It is widely found in quartz, granite, sandstone, slate, and sand [2]. Silica is primarily available in amorphous, crystalline, and cryptocrystalline forms, each with unique physical and chemical properties. The random orientation of the molecules in amorphous silica is reversed, and the arrangement of silica–oxygen tetrahedra in cryptocrystalline silica is irregular, whereas crystalline silica exhibits a fixed, repeating, polymerized silica–oxygen tetrahedral framework.

Silica has been reported to be released into the atmosphere primarily through crushing, grinding, or cutting materials containing silica. As it is odorless, colorless, non-irritating, and does not cause immediate health effects, it is easily overlooked [14]. Environmental exposure to silica is a growing concern not only during agricultural activities but also during natural dust storms and volcanic eruptions [15,16]. Similar to PM2.5, silica exposure varies among regions due to differences in development and industrial centers. As a result, occupational exposure limits (OELs) for silica dust have been established for each region, mainly including permissible concentration time-weighted average (PC-TWA). The data come from the United States Occupational Safety and Health Administration (OSHA), the European Chemicals Agency, and the National Health and Hygiene Commission of the People’s Republic of China (Table 2).

People working in environments with high concentrations of silica for long periods of time are prone to silicosis, a progressive, irreversible, and incurable chronic interstitial lung disease caused by the diffuse fibrosis of the lung tissue [17]. The severity and progression of the disease are limited by the dispersion of silica and the location and duration of residence [18]. In particular, free silica with a diameter of less than 5 μm is able to penetrate deeply into the fine bronchioles of the lungs and deposit in the alveoli. Meanwhile, it is easy to adsorb toxic substances on its surface, which makes it more likely to induce violent reactions in the body. The smaller the particle size, the faster the evolution of the disease process [19].

Correspondingly, silica particles with diameters less than 100 nm are known as nanosilica, a specific class of silica. Nanosilica has unique properties compared to its native form, with a larger surface-to-volume ratio, higher surface reactivity, and excellent dispersing ability [20]. These properties may account for the much higher potential toxicity of the nanoscale dimensions compared to their microscale counterparts [21]. Nanosilica has recently received the United States Food and Drug Administration (FDA) approval as an inorganic carrier in nanomedicine for a wide variety of practices due to its unique surface and physicochemical properties, including imaging, diagnostics, and therapeutics, leading to a significant increase in the industrial exposure of nanosilica during production, storage, transportation, and use [22]. At the same time, nanosilica can be easily mixed into the air via light air currents due to its low density, resulting in a significant increase in exposure for the general population [23]. Currently, nanosilica has become the second most produced nanomaterial on the global market with the development of nanosilica manufacturing [24].

Nanosilica is a metal oxide material consisting of a siloxane (Si-O-Si-O) core and an abundance of silanol (Si-OH) surface groups [25]. The silica backbone of nanosilica makes it stable in acidic conditions, temperature fluctuations, and organic solvents [26]. A uniform pore size, controllable particle size, large surface area, and easy surface modification with the presence of silanol groups, as well as good biocompatibility, endow it with the ability to be used as a carrier for other organic or inorganic materials [25,27]. It has been demonstrated that nanosilica can invade human organs through the respiratory tract, digestive tract, and skin. The smaller the nanosilica, the more efficiently it penetrates cells, suggesting the possibility of particle size-dependent toxicity [28]. Several studies have shown that the inhalation of nanosilica induces oxidative stress, endoplasmic reticulum (ER) stress, and apoptosis, further leading to inflammation and fibrosis [29].

In summary, exposure to PM2.5 is common in natural environments, while exposure to silica is mainly common in occupational environments. Exposure to nanosilica exists in both natural and occupational environments, leading to differences in the populations affected by these three respiratory dusts to some extent (Figure 1).

The surface-to-volume ratios and abilities to adsorb toxic substances of the different dusts are different due to the different particle sizes, contributing to the differences in the toxic effects. PM2.5 particles are prone to adsorb a variety of toxic substances on their surfaces, thus modulating inflammatory pathways and promoting oxidative stress and cell death. Currently, many regions have established 1 h average concentration limits for gaseous air pollutants such as SO_2_, NO_2_, CO, and O_3_. In addition, Japan has established 1 h average concentration limits for suspended particulate matter (SPM). However, there are no regulations on PM2.5 in this regard at present. Future research may pay more attention to the direction of the 1 h average concentration limit of PM2.5 to protect the health of the population. In contrast to PM2.5, which exists in the form of mixed particles containing a variety of chemicals, silica and nanosilica are specific chemicals that can absorb other toxic substances or bacteria or can exist alone without absorbing any additional substances. As is mentioned above, OELs for silica dust have been established in various regions. The implementation of OELs is part of a broader effort to protect workers from the significant health risks associated with silica exposure. However, workers who are exposed to silica experience increased breathing during labor, to the point where they inhale more silica than normal. Based on this, OSHA has set an action level for respirable crystalline silica at 25 µg/m^3^ over an 8 h TWA. Regrettably, several countries, China included, have not established a restriction like that yet. Future research may focus on this aspect to more fully protect workers from the hazards of silica dust. At the nanoscale, nanosilica is more likely to penetrate cell membranes and enter the cell interior than micrometer-sized silica, giving it a stronger biological activity and potential for toxicity. This may trigger more severe inflammatory responses and cellular damage. Unfortunately, safety assessments of the potential harmful effects of nanosilica have not yet been completed, and standards such as OELs are still lacking. Further research is needed to determine the exact biological mechanisms of nanosilica. 

## 3. Toxic Effects of PM2.5, Silica, and Nanosilica In Vivo

### 3.1. Animal Models

Current research has developed and implemented many animal models for studying the pathogenesis and molecular basis of in vivo damage caused by PM2.5, silica, and nanosilica to explore new therapeutic approaches (Table 3).

Mice and rats have become popular choices for experiments due to their diminutive size, ease of breeding and maintenance in large numbers, rapid life cycle, and convenient sample collection [21]. Among the common strains, C57BL/6 mice and Wistar rats are frequently used [30]. Zebrafish are also commonly used as animal models for studying the pathogenic effects of dust due to their ease of rearing, low maintenance costs, external fertilization and development, transparency in early life, and adaptability to chemical and genetic screening [31,32]. In addition, many mammalian species, such as pigs [33], dogs [34], river otters [35], camels [36], horses [37], and nonhuman primates [38], have been reported in past studies on the pathogenic effects of dust. Previous studies have shown that gender influences the development of pathological changes in the lungs, as well as the severity of lung damage. Nevertheless, researchers tend to select male animals in practice [39]. In addition, due to the inherent limitations of animal physiology, it is not possible to completely replicate and mimic the entire process of pathological changes in the human organism caused by prolonged exposure to dust particles of different sizes. To ensure the survival of animals, the aerosols and suspensions of PM2.5, silica, and nanosilica used for experiments must be sterilized before modeling. Concurrently, in the experiments, it is not possible to fully restore other minerals, metals, or bacterial materials on the surface of the silica and the chemical compositions attached to the surface of the PM2.5 and nanosilica that the patient was actually exposed to. Therefore, the exposure environment of animal models is different from the reality of exposed persons and, thus, the toxic effects of PM2.5, silica, and nanosilica cannot be fully assessed in animal models. Since all experimental animal models have their own shortcomings, there is a constant need to search for better models to accelerate the pace of basic research and to further aid in clinical treatment and drug development. In future studies, we should explore and develop dynamic animal models that are highly similar to human dust exposure situations and consider the combined effects of respiratory dust with other toxicants.

### 3.2. Organ Toxicity

Past studies have demonstrated that PM2.5 not only penetrates the gas exchange zone of the lungs but also crosses the respiratory barrier and enters the blood circulatory system. It then participates in metabolic regulation after entering the human body, ultimately leading to disorders in different organs and systems [40]. The lungs are considered to be the main organs affected by PM2.5. They are the initial site of PM2.5 deposition in the body after inhalation from the airways and are some of the main targets for the toxic effects of PM2.5. The inhalation of PM2.5 leads to airway inflammation, which further damages the alveolar structure and impairs the normal immune response of the lungs [41]. This eventually makes organisms susceptible to various respiratory infections. PM2.5 contains abundant organic compounds, such as polycyclic aromatic hydrocarbons (PAHs) and lipopolysaccharides, which can produce free radicals in the lungs. These compounds disrupt the rapid oxidative and antioxidant balance, causing damage to lung function and adverse effects on the lungs [42]. In addition, exposure to PM2.5 significantly alters the abundance, homogeneity, and composition of the lung microbiota, disrupting the levels of lung metabolites involved in multiple metabolic pathways and ultimately resulting in lung damage [43]. Emerging clinical and experimental evidence has examined and explained a range of extra-pulmonary toxicities of PM2.5. PM2.5 impairs the function of the cardiac autonomic nervous system (ANS) and leads to a decrease in heart rate variability (HRV), which has been suggested to be an independent risk factor for cardiovascular morbidity and mortality [44]. PM2.5 also triggers a range of pathophysiological responses, including increases in sympathetic tone, the modulation of basal systemic vascular tone, and endothelial and vascular dysfunction, resulting in an elevation of blood pressure and the progressive development of hypertension [45,46,47]. Exposure to PM2.5 has been reported to play a crucial etiological role in central nervous system (CNS) disorders. PM2.5, carrying toxic pollutants, can enter the respiratory tract and then enter the bloodstream through the alveoli. This can activate glial and microglial cell activity, leading to neuroinflammation and oxidative stress, ultimately resulting in damage to dopaminergic neurons and cerebrovascular injury [48,49,50]. As the skin is the largest and outermost organ of the human body, short-term exposure to PM2.5 can lead to specific dermatological disorders in different populations. PM2.5 adversely affects skin health through direct dermal uptake via percutaneous penetration and indirect dermal uptake through circulation, leading to the development and progression of skin disorders [51,52,53]. In addition, PM2.5 may be moved to the gastrointestinal tract via mucus cilia clearance or the ingestion of contaminated food and water, which may cause direct toxic effects on gastrointestinal epithelial cells and affect the composition of intestinal microbes, leading to increased intestinal permeability, altered intestinal immunity, oxidative stress, and inflammatory responses, ultimately damaging intestinal structures and decreasing intestinal function [54,55]. Studies have shown that PM2.5 impairs renal function mainly by inducing tubular atrophy, glomerulosclerosis, renal vascular injury, and interstitial fibrosis, which may involve oxidative stress, inflammatory response, cytotoxicity, and DNA damage [53,56]. Notably, the direct or indirect exposure of mothers to PM2.5 may result in multiple adverse birth outcomes. PM2.5 may reach the placenta, resulting in placental damage with effects on the fetal respiratory system, immune status, brain development, and cardiometabolic health [57]. Potential biological mechanisms include the direct placental translocation of PM2.5, placental and maternal systemic oxidative stress and inflammation, epigenetic changes, and potential endocrine effects that may affect long-term health [58]. In addition, recent animal studies have linked short- and long-term exposure to PM2.5 to adverse biological effects in organs such as the pancreas, bladder, and prostate [59,60,61].

The lungs, as the targets of silica action in the body, are under the greatest threat of toxicity and are susceptible to injury. The prolonged inhalation of free silica can lead to silicosis [62]. Immunization against silica is mainly achieved through the phagocytosis of lung macrophages [63]. Macrophages trigger extensive inflammatory cell infiltration in the organism’s lungs after ingesting silica and gradually transition to the fibrotic stage of persistent chronic inflammation and excessive tissue repair [64]. The current widely accepted pathogenesis of silicosis is as follows: (1) silica is identified and then phagocytosed by the alveolar macrophage (AM) via the scavenger receptor, which is the first critical defensive line against silica invasion; (2) dust-laden macrophages with scanty reticulin and collagen fibers die and release intracellular silica, which is then taken up by other AMs; (3) silicic acid produced by dissolved silica destroys the stability of the AM lysosomal membrane. Hydrolase released by the disrupted lysosome penetrates the cytoplasm and ultimately leads to AM death; (4) dead AMs can release a series of inflammatory factors, causing pulmonary inflammatory damage [65,66]. Correspondingly, AMs gather at the injured pulmonary tissue and stimulate the transformation of fibroblasts into myofibroblasts, leading to the excessive deposition of the extracellular matrix and eventual silicosis fibrosis [3,67]. After exposure to silica, AMs mediate particle clearance in rats, increasing the persistence of silica in the lungs. This leads to macrophage activation and the sustained release of chemokines and cytokines, further inducing genotoxicity, damage, and the proliferation of lung epithelial cells [68]. Previous studies have confirmed that silica, a potential risk factor for cardiovascular disease (CVD), may cross the pulmonary epithelium into the vascular bed, directly affecting the integrity of the vascular endothelium [69]. Silica can increase the secretion of proinflammatory cytokines and induce endothelial dysfunction through the stimulation of tumor necrosis factor α (TNF-α) expression and/or the increased recruitment of inflammatory cells [70,71]. This view is further supported by the observation of some investigators in a silica-induced pulmonary hypertension model. Mice exposed to silica showed signs of vascular remodeling, including the muscularization of the pulmonary arteries, vascular occlusion, and medial thickening [72]. Interestingly, inhalation and swallowing share the pharynx as a common structure, whereas they may not even be coordinated in some individuals [73]. Accordingly, silica may cause adverse health damage to extra-pulmonary organs due to this physiological feature. Silica ingested in the workplace may stimulate mucosal proliferation during retention in the gastrointestinal tract by inducing DNA damage or the increased secretion of cytokines and growth factors. This, in turn, causes chronic localized damage and inflammation, further inducing, stimulating, or promoting tissue carcinogenesis [74]. In addition, silica may induce autoimmune diseases through direct nephrotoxicity or indirect toxicity, primarily by causing the production of autoantibodies against nuclear and other self-antigens deposited in the kidneys. This can lead to autoimmune diseases and kidney damage, including renal tubular epithelial and mesenchymal inflammation, fibrillar nephropathy, and glomerulonephritis [75,76,77]. Moreover, the toxic effects of silica on the skin, bones, and liver have been reported but the specific mechanisms are largely unknown. Further comprehensive investigations are needed in order to fully comprehend the exposure hazards and the specific toxicological characteristics.

The multi-organ toxicity of nanosilica has been a hot topic in recent years and has been studied comprehensively. Nanosilica, with a larger surface-to-volume ratio and higher surface reactivity than silica, is more likely to enter the human body through respiration, ingestion, and touch, and to further distribute to various tissues through the circulatory system [26]. By establishing animal models through the direct injection of nanosilica into lung tissue, intratracheal perfusion, or intranasal perfusion, researchers have found that the in vivo administration of nanosilica induces effects such as acute lung inflammation, pulmonary fibrosis, alveolar injury, and granulomatous nodule formation [78,79]. The collection and testing of lung tissue or bronchoalveolar lavage fluid (BALF) revealed that the toxicity induced by nanosilica was mainly associated with increased lipid peroxidation, the high expression of cytokines, the accumulation of transforming growth factor 1 (TGF-β1) on the corona of the nanosilica surface, increased mucus secretion, the disruption of connective tissues, the perturbation of collagen metabolism in the respiratory interstitium, and airway inflammation [80,81,82,83,84]. After endotracheal instillation, nanosilica can enter the circulation through the alveolar–capillary barrier, increase the levels of inflammation-related high-sensitivity C-reactive protein (Hs-CRP) and cytokines in the blood, and significantly increase the secretion of vascular cell adhesion molecule-l (VCAM-1), causing endothelial dysfunction and accelerating thrombosis [85,86]. Meanwhile, nanosilica contributes to coronary artery thrombosis by interfering with cardiomyocyte ion channels and transmembrane potential, leading to coronary endothelial injury and coronary coagulation. This further exacerbates atherosclerotic lesions through the ER stress-mediated upregulation of platelet glycoprotein 4 (CD36) expression in macrophages [87,88]. Numerous studies in vitro have shown that the mechanism of neurotoxicity of nanosilica is mainly related to the promotion of oxidative stress and the stimulation of proinflammatory responses [89]. Additionally, nanosilica can bind to specific proteins, such as apolipoproteins, to form protein crowns. This allows them to cross the blood–brain barrier and reach the central nervous system through the nasal epithelium, potentially causing neurotoxicity [90,91]. In microglia, nanosilica induces the expression of inflammation-related genes by promoting the secretion of TNF-α, interleukin-1β (IL-1β), and interleukin-6 (IL-6) [92]. Meanwhile, nanosilica has been shown to increase superoxide dismutase and catalase activities, as well as IL-1β levels, leading to oxidative stress, inflammation, and DNA damage in the brain [93]. Increasing evidence supports a link between exposure to nanosilica and hepatic adverse effects. As a secondary exposure site, the liver preferentially accumulates nanosilica compared to other organs, with more than 90% of translocated nanosilica remaining in the liver [94]. Increased reactive oxygen species (ROS) is the molecular initiating event in the hepatotoxicity of nanosilica, which then leads to critical events at the molecular and cellular levels, including oxidative stress, ER stress, lysosomal disruption, mitochondrial dysfunction, hepatic parenchymal cell dysfunction, and hepatic macrophage dysfunction. These events ultimately progress to adverse outcomes such as hepatic dysfunction and liver fibrosis [95]. In order to assess the in vivo toxicity of nanosilica in the gastrointestinal tract, the most commonly used method is to administer nanosilica orally or intravenously to induce acute toxicity [96]. Studies have confirmed that nanosilica significantly alters the diversity of intestinal flora in mice, leading to a microecological imbalance and the disruption of intestinal mucosal epithelial structure. This eventually induces the infiltration of inflammatory cells and causes intestinal inflammation. Meanwhile, nanosilica may affect the expression of metabolites involved in a series of metabolic pathways, including pyrimidine metabolism, purine metabolism, central carbon metabolism in cancer, protein digestion and absorption, and mineral absorption, exacerbating intestinal inflammation [97]. The inhalation of nanosilica may cause renal interstitial inflammation, which can lead to tubular damage and reduced renal function, ultimately resulting in chronic kidney injury [98]. It is hypothesized that the toxic effect is largely triggered by the increased production of ROS and the peroxidation of unsaturated lipids [99]. Studies have confirmed that nanosilica is able to cross the placental barrier and reach the fetal bloodstream. It is significantly metabolized and absorbed by developing organs, including the fetal brain, stomach, thymus, liver, and lungs, leading to fetal systemic toxicity and maternal pregnancy complications [100].

In summary, respiratory dusts with different particle sizes can cause toxicity to various organs and systems in the human body (Table 4).

Among them, PM2.5 and nanosilica have smaller particle sizes and are able to cross the blood–brain and placental barriers, which can adversely affect the nervous system and fetal development. Furthermore, the greater surface-to-volume ratios of PM2.5 and nanosilica result in an enhanced ability to adsorb and carry toxic substances in the atmosphere, such as heavy metals, organic pollutants, and carcinogens. Once inhaled by the human body, these toxic substances may be deposited in the respiratory tract, enter the alveoli, and be absorbed into the bloodstream. These toxic substances may promote the release of inflammatory mediators and the activation of inflammatory cells, which further leads to a series of adverse physiological effects such as cellular damage and tissue inflammation, ultimately damaging multiple organs. Since nanoscale particles can more easily penetrate cell membranes and enter the cell interior, nanosilica may trigger a more intense inflammatory response within the cell, thereby exacerbating the development and progression of various diseases. In contrast, since silica has larger particles and lower dispersion, there are fewer reports of its adverse health effects on the nervous system and fetus through the blood–brain and placental barriers. However, under prolonged exposure, silica can still enter the bloodstream through gas exchange and cause serious effects on multiple systems in the human body. From the current study, silica has a relatively weaker inflammatory effect than PM2.5 and nanosilica, while it has a stronger fibrogenic ability, mainly causing silicosis. Overall, respiratory dusts with different particle sizes may have a wide range of effects on human health, highlighting the necessity for continued in-depth studies to enhance our understanding of their toxic effects and health implications.

## 4. The Cytotoxicity of PM2.5, Silica, and Nanosilica In Vitro

### 4.1. Cellular Models

Although in vivo models based on experimental animals may provide data that can be directly extrapolated to humans, the translation of these models to human subjects is highly unpredictable. The scientific community has expressed concerns about the clinical validity of experimental animals, as well as ethical issues [101]. Toxicologists are working to develop in vitro models to evaluate the inhalation toxicity of PM2.5, silica, and nanosilica. The application of in vitro models is relatively simple and rapid compared to in vivo models. Currently, cultured human lung adenocarcinoma epithelial cells, particularly A549 cells, are most commonly used as in vitro models for assessing the inhalation toxicity of dust. The ultrastructural features of the A549 cells are similar to those of type II lung cells in situ. Not only are they relatively easy to cultivate but they are also an established model for studying alveolar type II cell responses, allowing a large number of comparative experiments to be performed in a short time [102]. However, A549 cells are cancerous, and proteomic analyses have shown fundamental differences in protein patterns between A549 cells and normal human bronchial epithelial cells (16HBE cells) [103]. Although these differences may be caused by the different origins of the cell types, it can be expected that A549 cells may not reflect the response of normal lung tissue to dust exposure. Interestingly, when 16HBE cells and A549 cells were used as in vitro models to evaluate the toxicity of nanosilica on lung cells, their sensitivity to nanosilica was different, with 16HBE cells being more sensitive [104]. This may provide a good option for future studies in related aspects. In addition, human leukemia monocytic cells (THP-1 cells) are commonly used as replacements for human primary macrophages. They are widely used to study the function of monocytes/macrophages and relevant mechanisms after exposure to PM2.5, silica, and nanosilica. THP-1 is an immortalized cell line that can be cultured in vitro for up to 25 generations (approximately 3 months) without altering cell sensitivity and activity [105]. More importantly, the homogeneous genetic background of THP-1 minimizes the degree of cellular phenotypic variability, which promotes the reproducibility of results, making it a suitable cellular model for studying dust toxicity [106,107]. The robust research models not only contribute to the accuracy of predictions but are also essential for understanding the complete characterization, the processes upon contact with living systems, and the possible toxicological effects of PM2.5, silica, and nanosilica. Therefore, in order to increase the predictability of the highly dynamic and multifactorial nature of this toxicity, future in vitro studies should carefully select and develop appropriate models to assess their individual and combined inhalation toxicity.

### 4.2. Cytotoxicity

The cytotoxicity of dust is a multifactorial process, with particle size and surface area being the clear determinants [23]. This cytotoxicity, which may affect cell growth and differentiation, can cause abnormal cell proliferation and death. It needs to be verified by more studies. Currently, in vitro toxicity assays have been used to investigate the cytotoxicity of PM2.5, silica, and nanosilica, as well as the associated mechanisms.

The airways and lungs are the initial sites of the entry and deposition of PM2.5, making them the primary targets. PM2.5 inhaled is deposited on the airway, lung bronchioles, and alveolar surfaces, and then internalized into lung cells such as epithelial cells and alveolar macrophages [108]. Thereafter, PM2.5 triggers oxidative stress and impairs normal cellular functions, eventually inducing cell death through different mechanisms. Studies have shown that the exposure level and duration of PM2.5 affects macrophage survival. High doses and prolonged exposure cause elevated levels of TNF-α and C-reactive protein (CRP) in macrophages, which stimulate signals to move from extracellular to intracellular compartments, activate the nuclear factor kappa-B (NF-κB) signaling pathway, and induce a cascade of inflammation [109]. In human normal lung epithelial cells (Beas-2b cells), PM2.5 has been reported to cause a NOX4/Nrf2 redox imbalance and trigger excessive ROS production, ultimately leading to oxidative stress, excessive mitochondrial autophagy, and mitochondrial damage in a dose-dependent manner [110]. In A549 cells, PM2.5 activated the Keap1/Nrf2 pathway, leading to excessive intracellular ROS production and significant cytotoxicity in a dose-dependent manner [111]. In HBE cells, PM2.5 significantly inhibited viability, caused DNA damage, and induced apoptosis in a concentration-dependent manner when compared to the normal control. The higher the concentration, the greater the cell damage [112]. The PM2.5 exposure-induced disruption of cellular homeostasis and cell death is one of the major mechanisms in the pathogenesis of neurodegenerative diseases. In mouse neuroblastoma Neuro-2a cells (N2a cells), PM2.5 exposure produced cytotoxicity and triggered cell death through the activation of the p62-Keap1-Nrf2 pathway and autophagy–lysosomal dysfunction in a concentration-dependent manner. This was related to an increase in intracellular ROS and disruption of the physiological functions of macromolecules such as proteins, nucleic acids, and lipids [113]. When human keratinocyte-forming cells (HaCaT cells) were exposed to PM2.5, the PM2.5 could stimulate HaCaT cells to release a series of cytokines and proinflammatory cytokines. These further stimulated epithelial cells, fibroblasts, and endothelial cells to secrete cytokines and adhesion molecules (such as IL-8, IL-2, and IL-1), suggesting that PM2.5 might increase the risk of eczema and other skin diseases that depend on inflammatory activity through oxidative stress production [114]. The effects of PM2.5 on renal cells in vitro have been focused on human embryonic renal cells (HEK-293 cells), podocytes, human renal cortical proximal tubule epithelial cells (HK-2 cells), mouse monocyte macrophages (RAW264.7 cells), and vascular endothelial cells. PM2.5 exposure reduced the cell viability of renal-derived cell lines, promoted oxidative stress and inflammatory responses, and induced cytotoxicity in a time- and concentration-dependent manner [115,116]. When human chorionic trophoblast cells (HTR-8/SVneo cells) were exposed to different concentrations of PM2.5, the steroidogenesis in trophoblast cells or other upstream mechanisms may have been interfered with, causing an increase in the production of IL-6 and a decrease in the production of hCG in the cells. This, in turn, resulted in a decrease in cell viability and ER stress [117]. PM2.5 was also found to specifically target mitochondria in trophoblast cells, leading to mitochondrial structural damage and dysfunction [117]. In human embryonic stem cells (EBf-H9 cells), exposure to PM2.5 reduced cell viability, increased levels of lipid peroxidation, and induced the production of ROS, resulting in oxidative stress and damage to the cells [118].

Silica has piezoelectric properties; that is, the crystals acquire electrode properties under applied pressure [119]. It is theorized that these piezoelectric characteristics may play a role in the pathophysiology of silica-related illness through the generation of oxygen free radicals produced on the cleaved surfaces of silica molecules and as a result of silica-damaged cells [119]. Silanol groups present on the surface of silica particles are capable of forming hydrogen bonds with oxygen and nitrogen groups found in biologic cell membranes, thereby providing a direct means of cytotoxicity as the particles react with resident cells, leading to the lipid peroxidation of the cell membrane [120]. Macrophages and fibroblasts are the main effector cells of silica-induced pulmonary fibrosis. In vitro experiments using mouse alveolar macrophages (MH-S) as a model found that sustained silica stimulation resulted in the excessive production of mitochondrial reactive oxygen species (mtROS) and increased cellular oxidative stress. One of the adverse consequences was an increase in inflammatory factors and cytokines, which further led to mitochondrial swelling, the rupture of the mitochondrial membrane, mitochondrial apoptosis, a decrease in mitochondrial number, the inhibition of cell proliferation, and the induction of apoptosis [121]. As reported, silica increased the levels of collagen I (COL I) and α-smooth muscle actin (α-SMA) in human lung fibroblasts (MRC-5 cells) and activated ferroptosis in HBE cells [122]. Notably, SiO_2_-stimulated A549 and HBE cells showed an increase in the expression of α-SMA and waveforming protein and a decrease in the expression of E-calmodulin in a dose-dependent manner [123]. This suggests that the cells underwent an epithelial–mesenchymal transition (EMT) process. EMT could activate transcription factors, secrete pro-fibrotic cell-surface proteins and cytokines, and increase extracellular matrix (ECM) accumulation [124]. Moreover, epithelial cells could participate in myofibroblast development through the EMT process. Silica treatment also inhibited cell viability in A549 and THP-1 cells, induced the secretion of inflammatory factors, and triggered an inflammatory response. Meanwhile, silica stimulated fibroblast activation in MRC-5 cells via an Adenosine 5′-monophosphate (AMP)-activated protein kinase (AMPK)-dependent pathway. This resulted in cell apoptosis, high levels of ROS, the decreased antioxidant activity of glutathione (GSH), and increased levels of the lipid peroxidation of MDA, ultimately leading to increased cellular oxidative stress [123]. In Beas-2b cells and primary human bronchial epithelial cells (PBECs), crystalline silica induced canonical Wingless-type MMTV-integration site (Wnt) signaling (β-catenin) and decreased non-canonical (Wnt5A) signaling, suggesting that Wnt signaling and cross-talk with other pathways (e.g., Notch signaling), might contribute to proliferative, fibrogenic, and inflammatory responses to silica in lung epithelial cells [125].

A large number of experiments have shown that the cytotoxicity of nanosilica appears to be highly dependent on size, dose, cell type, and the route of administration [20]. That is why some studies report no significant toxicity of nanosilica, while others report the greater toxicity of smaller-sized nanosilica (<50 nm), and still others report the greater toxicity of larger-sized nanosilica (>50 nm) [126,127]. For example, in human primary bronchial epithelial cells, the number of upregulated genes in response to smaller diameter nanosilica exposure was higher. These genes included those involved in inflammatory responses, apoptosis, oxidative stress, and DNA damage repair. In these cells, 41 nm nanosilica elicited a greater cytotoxic response than 61 nm nanosilica, and both sizes of nanoparticles were more cytotoxic than micrometer-sized silica particles. Apoptosis was observed only after treatment with nanoparticles [128]. This was further supported by the fact that A549 cells and THP-1 cells exposed to 60 nm amorphous nanosilica had a significantly higher mortality rate than larger silica particles [129]. A previous study found that the cell viability of A549 cells exposed to nanosilica (15 and 46 nm) was reduced in a time- and dose-dependent manner, and the cytotoxicity of 15 nm nanosilica was greater. Exposure to 15 nm nanosilica generated oxidative stress in A549 cells, as reflected by reduced GSH levels, the elevated production of MDA, and lactate dehydrogenase (LDH) leakage [130]. This is indicative of lipid peroxidation and membrane damage, respectively. Nanosilica could also lead to necrosis of endothelial cells and high levels of oxidative stress. Different types of human endothelial cells have been widely used for in vitro studies on the toxicity of nanosilica. In human umbilical vein endothelial cells (HUVECs), nanosilica of different particle sizes exhibited a variety of potential toxicities. HUVECs treated with 310 nm nanosilica resulted in decreased mitochondrial activity and increased membrane leakage, while treatment with 60 nm nanosilica induced cytotoxicity through oxidative stress, resulting in mitochondrial dysfunction, redox imbalance, and lipid peroxidation [89,131,132]. Meanwhile, both 20 nm and 100 nm nanosilica inhibited HUVEC migration and tube formation, as well as causing significant calcium mobilization [133]. In microglia cells (N9 cells), which represent CNS-dwelling macrophages, nanosilica exhibited particle-size-dependent toxicity, leading to mitochondrial ROS production, the expression of proinflammatory cytokines, and gasdermin D (GSDMD) cleavage and pyroptosis [134]. In human neuroblastoma cells (SH-SY5Y cells), nanosilica decreased cell viability, caused damage to the neural cell membrane, and increased the intracellular Ca^2+^ level. Moreover, exposure to nanosilica also led to the loss of mitochondrial membrane potential (MMP), damage to mitochondria, and the elevated expression of apoptotic protease-activating factor-1 (Apaf1) and cytochrome C (Cyt C). This triggered caspase cascades of caspase-9 and caspase-3, ultimately resulting in neural apoptosis [135]. The toxicity of nanosilica in the liver has been well studied using various cell lines in vitro. In human liver cancer cells (HepG2 cells), exposure to 15 nm nanosilica caused alterations in global metabolomics, specifically the depletion of glutathione, NADPH oxidase-mediated ROS formation, and alterations in the antioxidant enzyme system [136]. Another related study found that nanosilica-induced apoptosis in HepG2 cells may be mediated and regulated through the p53, Bax/Bcl-2, and cysteine asparaginase pathways [137]. Similarly, the toxicity of 20 nm nanosilica on human normal liver cells (HL-7702 cells) and rat normal liver cells (BRL-3A cells) was associated with the three pathways mentioned above and mediated through oxidative stress [138]. Human intestinal epithelial cells (HT-29 cells and Caco-2 cells) are typical cells used for in vitro studies targeting the gut. A simulated gastrointestinal epithelial model using Caco-2 and HT29-methothexate (HT29-MTX) co-cultures was used to study the acute and chronic toxicity of nanosilica. It was found that the nanoparticles significantly affected the absorption of iron (Fe), zinc (Zn), glucose, and lipids, as well as barrier function. They also decreased the number of intestinal microvilli and significantly increased the activity of the brush border membrane enzyme, intestinal alkaline phosphatase. Furthermore, they altered the expression levels of nutrient transport proteins and induced the production of ROS and proinflammatory signaling [139]. Another study showed that the effect of nanosilica was not mediated by oxidative stress, but rather by interference with the MAPK/ERK1/2 and Nrf2/ARE signaling pathways [140].

The cytotoxic effects of PM2.5, silica, and nanosilica may be realized through several pathways [23]:Cause oxidative stress by increasing levels of lipid peroxidation through increased ROS, which can lead to intracellular oxidative damage and affect normal cellular function.Destroy mitochondria and disrupt their metabolism, causing mitochondrial autophagy and damage, which lead to an imbalance in cell energy and, ultimately, cell death.Disturb the formation of lysosomes, thereby impeding the autophagy and degradation of macromolecules, leading to the accumulation of intracellular waste and triggering apoptosis.Disrupt transcription and damage DNA, thereby accelerating mutagenesis, which further leads to abnormal cell proliferation and tumorigenesis.Disturb the normal mechanism of cellular metabolism, activate the synthesis of inflammatory mediators, and induce a series of inflammatory responses. These inflammatory responses may trigger tissue damage and the development of inflammatory diseases.Destroy the cell membrane via perforation, causing an imbalance between the internal and external environments of the cell, which in turn affects the normal function and survival of the cell.

The combined effects of these pathways may lead to cell death, further contributing to tissue damage, inflammatory responses, and potentially disease development.

Exposure time and concentration are key factors in dust cytotoxicity and can even influence the severity of the effect of particle size. Previous studies have suggested that the toxicity of nanoscale and microscale dusts varies significantly at concentrations of ≤50 μg/mL and concentrations of ≥100 μg/mL [141].

We also found that in dust toxicity studies based on in vitro models, the overall dosing concentrations of silica and PM2.5 were significantly higher than those of nanosilica. This may represent the fact that nanoscale dust causes a more dramatic toxic response at the same concentration. In addition, in vitro studies further confirmed the cytotoxicity of PM2.5 and nanosilica in multiple systems while the cytotoxicity studies of silica were limited to lung cells, similar to in vivo studies (Table 5).

The various mechanisms may be the main factors contributing to their different toxic effects in vivo and open up more possibilities for the consequences of combined exposures (Figure 2).

The mechanisms of toxic effects of different dusts are still incompletely studied, which may make the further analysis of their synergistic effects more challenging. Therefore, an in-depth study of the mechanisms of cytotoxicity of these dusts is of great importance.

## 5. The Population Health Effects of PM2.5, Silica, and Nanosilica

Epidemiologic studies can be designed to focus on short- or long-term health outcomes, depending on the characteristics of the exposure. Cumulative respiratory exposure to PM2.5 and silica has been identified as a major factor associated with the development of pulmonary pathology and is influenced by the duration and intensity of exposure [3,142]. Previous epidemiologic studies related to PM2.5, silica, and nanosilica have primarily focused on long-term exposures, comparing spatial variations in dust concentrations and the occurrence of health events. Epidemiologic studies assessing the effect of long-term dust exposure on mortality have primarily relied on cohort studies, in which a group of individuals is followed longitudinally and individual exposure doses to dust and the occurrence of death are compared [143,144]. The integrated exposure–response (IER) estimated from epidemiologic cohort studies is a key component in assessing the global burden of disease from PM2.5 and silica [145].

The World Health Organization (WHO) air quality guidelines estimate that reducing the annual average PM2.5 concentration from 35 μg/m^3^ to the recommended level of 10 μg/m^3^ could reduce air pollution-related deaths by approximately 15%. Recent findings suggest the importance of examining the biological responses and possible underlying mechanisms associated with PM2.5 in the context of cardiopulmonary outcomes [146]. The adverse effects of PM2.5 pollution on human health have been well studied, with a large number of studies demonstrating clear associations between long-term exposure to PM2.5 and public health risks, such as respiratory, cardiovascular, and neurological diseases, as well as adverse perinatal outcomes and birth defects [147]. Results from seven large European cohorts showed that low concentrations of PM2.5 were positively associated with non-accidental and cause-specific mortality [148]. In a large U.S. study based on data from the Surveillance, Epidemiology, and End Results program, a 10 μg/m^3^ increase in county-level PM2.5 estimates was associated with a 19% increased risk of lung cancer [11]. A UK population-based prospective cohort study demonstrated that long-term exposure to PM2.5 was associated with an increased risk of depression and anxiety and that reducing exposure could potentially reduce the disease burden of depression and anxiety [149].

Silica exposure is one of the most common occupational exposures. In the United States, more than 2 million workers in the construction industry and 300,000 workers in other industries, including brick manufacturing, foundries, and hydraulic fracturing, are occupationally exposed to silica [150]. It is estimated that about 3 to 5 million workers in Europe are exposed [151]. Meanwhile, a cross-sectional survey of the Australian working population shows that 6.6% of Australian workers are exposed to silica, and 3.7% are highly exposed while performing work tasks [152]. Silica exposure is also very common in low- and middle-income countries. In India, approximately 3 million workers are at high risk of silica exposure. Out of these, 1.7 million are involved in mining or quarrying activities, 600,000 are involved in the manufacturing of non-metallic products (such as refractory products, structural clay, glass, and mica), and 700,000 are involved in the metal industry. In addition, there are approximately 5.3 million construction workers at risk for silica exposure [153]. Silica has been classified as a Group 1 carcinogen by the International Agency for Research on Cancer (IARC) since 1997 [154]. In recent years, the adverse health effects caused by silica have been extensively studied. Corresponding epidemiologic evidence suggests an association between silica exposure and increased morbidity and mortality due to the exacerbation of pre-existing cardiopulmonary diseases [155]. In general, acceptable exposure limits for silica range from 0.05 mg/m^3^ to 0.1 mg/m^3^. However, later studies have demonstrated that even workers with exposure levels equal to a 0.05 mg/m^3^ time-weighted average (TWA) over a 40- to 45-year working career are still at significant risk (at least a 1 in 100 possibility) of developing radiographic silicosis [156]. Pooled data from 10 international cohort studies suggest that sustained exposure to silica at a level of 0.1 mg/m^3^ over a 45-year period is associated with a 1.1–1.7% increase in the lifetime risk of silicosis [157]. Data from a large number of epidemiological investigations have shown that prolonged exposure to silica in the work environment primarily increases the risk of silicosis in workers. This exposure can also contribute to other complications such as lung cancer and emphysema, leaving many young workers incapacitated and, in some cases, even dead [158]. In a cohort study of the total Danish working population, researchers found an exposure-dependent association between occupational exposure to silica and autoimmune rheumatic diseases, most pronounced in systemic sclerosis and rheumatoid arthritis [159].

To the best of our knowledge, there is a lack of relevant epidemiological studies investigating the health effects of occupational exposure to manufactured nanosilica through physiological inhalation. Large cohort analyses are not suitable for studying the population health effects of nanosilica due to the time-consuming and costly nature of the study, as well as the technical difficulties of measuring the dose of inhaled nanoparticles in patients. As a result, the potential health complications associated with nanosilica remain largely unknown, and further studies are needed to characterize the exposure characteristics and corresponding health hazards of individuals exposed to nanosilica in various application scenarios. Nevertheless, there are still several epidemiological studies that suggest that increased levels of environmental nanosilica are associated with increased human morbidity, mortality, and the incidence of cardiovascular disease [160,161].

Epidemiological studies conducted in different regions have confirmed the association between exposure to respiratory dust and respiratory and cardiovascular health risks. However, as most of the evidence is obtained from studies in individual cities, regions, or countries, there is the potential for publication bias. Therefore, there are challenges in comparing these results and estimating the combined effects.

## 6. Combined Toxicity

As exposure opportunities to PM2.5, silica, and nanosilica increase in daily life, their interactions and the resulting combined toxic effects deserve further attention. However, the available studies are limited in terms of the combined toxicity of PM2.5, silica, and nanosilica, and the underlying mechanisms remain unclear.

The rapid growth of hydraulic fracturing for oil and gas extraction in the United States has resulted in a marked increase in the number of active frac sand mines, processing plants, and railroad transfer stations. Ambient particulate monitors found significantly higher concentrations of PM2.5 and silica in emitted air pollutants, placing operational workers at a significantly higher risk of simultaneous exposure to PM2.5 and silica [162]. An analysis of ambient atmospheric particles in the human lung airway retention revealed that both respirable silica and PM2.5 tended to accumulate in the airways. This implies that simultaneous or sequential exposure to silica and PM2.5 may result in more dramatic toxic effects than exposure alone [163]. Spirometry examinations of two groups of Swedish soldiers exposed to Desert Storm, which contained silica and PM2.5, found that their forced expiratory volume in one second (FEV1) and forced vital capacity (FVC) were lower than before exposure [155]. However, as other confounding factors were not controlled in the current study, the results obtained may be biased. Nanosilica is one of the major surface components of PM2.5, so the combined exposure to nanosilica and PM2.5 is almost inevitable [164]. It has been demonstrated that both PM2.5 and nanosilica can cause toxic effects on multiple organs and systems throughout the body via blood circulation. The toxic effects of these substances are largely similar, mainly through oxidative stress and inflammatory response. Moreover, PM2.5 and nanosilica can cross both the blood–brain and placental barriers. Nanosilica may even target the brain via the systemic pathway and accumulate in the brain, further affecting the nervous system and the fetus. This makes the consequences of combined exposure even more unpredictable. Although there are no reports of silica crossing the blood–brain and placental barriers, it can still cause multi-system toxic effects through systemic lymph. By thoroughly investigating the mechanisms of toxic effects of PM2.5, silica, and nanosilica, we found that as respiratory dusts, all three share a common target—the lungs. Additionally, all of them can cause toxicity to other organs. It can be hypothesized that combined exposure is likely to increase the toxic effects and the body’s compensatory burden. Nevertheless, there is a lack of relevant studies. What are the toxic effects of simultaneous or sequential exposure to ambient PM2.5, silica, and nanosilica? Do combined exposures produce synergistic effects? What are the specific mechanisms? These questions deserve further exploration.

## 7. Conclusions

In this review, we described the sources and characteristics of three different particle sizes of dust: PM2.5, silica, and nanosilica. Based on their respective characteristics, we further explored the main toxicity induced by PM2.5, silica, and nanosilica in vivo and in vitro. Furthermore, we evaluated the health implications of respiratory dust on the human body and proposed potential synergistic effects, taking into consideration current studies. Overall, the severity of organ damage and cytotoxicity induced by PM2.5, silica, and nanosilica varies and is influenced by multiple factors, including dust type, size, dose, and surface composition. Other key factors, such as cell type, cell status, organ distribution, animal status, and the duration of exposure, also play a role and are subject to individual differences in cells and in vivo. The toxicity mechanisms of PM2.5, silica, and nanosilica include oxidative stress, inflammation, DNA damage, and metabolic disorders. Currently, the reported injuries in humans are focused on respiratory diseases, cardiovascular diseases, neurological diseases, and adverse pregnancy outcomes. However, studies on the related signaling pathways are still relatively incomplete.

We hope that toxicity studies will continue to delve deeper and improve the upstream and downstream signaling regulatory mechanisms. There are still challenges in fully understanding the toxicity of dust particles of different sizes in humans. Future studies should carefully determine the dose that induces toxic responses in humans and consider it in conjunction with clinical and epidemiological data. Meanwhile, mechanism studies need to consider complex physiological systems. More importantly, PM2.5, silica, and nanosilica inevitably coexist in the atmosphere, emphasizing the importance of combined toxicity studies. Besides enhancing toxicity studies, we suggest implementing more policies aimed at decreasing environmental levels of PM2.5, silica, and nanosilica, specifically by revising exposure limits to minimize overall population exposure.

## Figures and Tables

**Figure 1 ijms-25-07654-f001:**
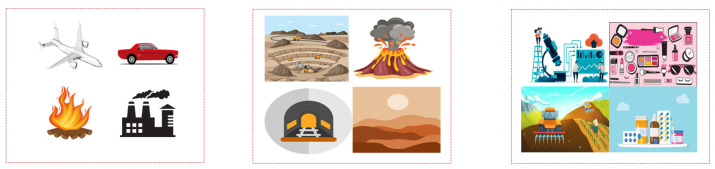
Multiple sources of PM2.5, silica, and nanosilica in the environment. Vehicle emissions, factory emissions, and fuel combustion increase the concentration of PM2.5 in the atmosphere; dust from mine blasting, tunnel excavation, dust storms, and volcanic eruptions increases the concentration of silica in the environment; the use of nanosilica in laboratories, cosmetics, agriculture, and pharmaceuticals increases the risk of exposure for those involved.

**Figure 2 ijms-25-07654-f002:**
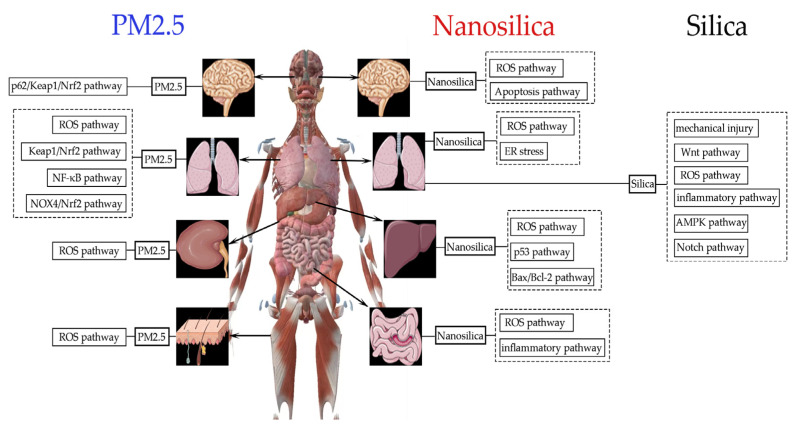
Mechanisms involved in the damage caused by PM2.5, silica, and nanosilica to different parts of the human body. The p62/Keap1/Nrf2 pathway is responsible for the effects of PM2.5 on brain cells, while the ROS, Keap1/Nrf2, NF-κB, and NOX4/Nrf2 pathways are involved in its impact on lung cells. Kidney cells and keratinocytes are affected by PM2.5 through the ROS pathway. Normally, nanosilica damages cells from multiple organs through ROS, apoptosis, p53, Bax/Bcl-2, and inflammatory pathways, while silica mainly damages lung cells through Wnt, ROS, AMPK, Notch, and inflammatory pathways, as well as mechanical injury.

**Table 1 ijms-25-07654-t001:** Environmental standards for PM2.5 in different regions.

Political Bodies	United States		European Union	Japan	Korea	China	
	Primary *	Secondary **				Primary ***	Secondary ****
Daily (µg/m^3^)	35	35	-	35	35	35	75
Annual (µg/m^3^)	9	15	25	15	15	15	35

* The primary standard for PM2.5 in the United States is for health protection; ** the secondary standard for PM2.5 in the United States is for public welfare protection; *** the primary standard for PM2.5 in China applies to nature reserves, scenic spots, and other areas requiring special protection; **** the secondary standard in China applies to residential areas, mixed commercial–transportation residential areas, cultural areas, industrial areas, and rural areas.

**Table 2 ijms-25-07654-t002:** Occupational exposure limits for silica dust in different regions.

PC TWA (mg/m³)	United States	European Union	China		
			10% ≤ Free SiO_2_ * ≤ 50%	50% < Free SiO_2_ ≤ 80%	Free SiO_2_ > 80%
Total dust	-	-	1	0.7	0.5
Respirable dust	0.05	0.1	0.7	0.3	0.2

* The content of free SiO_2_ in dust.

**Table 3 ijms-25-07654-t003:** Advantages and disadvantages of several animal models.

Animal Models	Advantages	Disadvantages
Mice	Diminutive size, ease of breeding and maintenance in large numbers, rapid life cycle, and convenient sample collection	Physical characteristics are not exactly the same as humans; the reproducibility of experimental results is poor; limited expression of human disease genes
Rats	Medium size, docile, high blood volume, and fertility	
Zebrafish	Ease of rearing, low maintenance costs, external fertilization and development, transparency of early life, and adaptability to chemical and genetic screening	
Pigs	Similar to humans in anatomical size and structure, physiology, immunity, and genome	
Dogs	Ease of compliance and high test fit	

**Table 4 ijms-25-07654-t004:** In vivo toxicity induced by PM2.5, silica, and nanosilica.

Animal Models	Types of Dust	Treatment	Main Results	Reference
C57BL/6N mice	PM2.5	1.8, 5.4, and 16.2 mg/kg PM2.5 through intratracheal instillation on day 1, day 4, and day 7	Exposure to PM2.5 significantly altered the abundance, homogeneity, and composition of the lung microbiota	[43]
C57BL/6J mice	PM2.5	Exposed to concentrated ambient PM2.5 for 6 months	Long-term PM2.5 exposure increased blood pressure through sympathetic nervous system activation	[45]
*ApoE*^−/−^ mice	PM2.5	Instilled with PM2.5 at a dose of 4 mg/kg	PM2.5 exposure directly activated inducible nitric oxide synthase (iNOS) in the lungs to produce excess nitric oxide (NO) as the initiating factor of vascular dysfunction	[46]
C57BL/6JGpt mice	Crystalline silica	30 mg/kg silica through intratracheal instillation	Crystalline silica induced pulmonary inflammation and fibrosis in mice	[64]
C57BL6 mice	Crystalline silica	Intratracheally injected with crystalline silica at doses of 0.2, 0.3, and 0.4 g/kg	Silica promoted the damage of the pulmonary vasculature through mechanisms that might involve endothelial dysfunction, inflammation, and vascular remodeling	[72]
Male Wistar rats	Synthetic silica nanoparticles	Instilled intratracheally with 1 mL of saline containing 6.25, 12.5, and 25.0 mg of nanosilica for 30 d	The nanosilica resulted in pulmonary fibrosis by means of increased lipid peroxidation and the high expression of cytokines	[78]
Female C57BL/6 mice	Nanosilica	Instillation at doses of 0.1 mg/kg and 0.05 mg/kg on days 2–4	The respiratory tract exposure to nanosilica led to inflammatory responses, including an increase in the number of inflammatory cells and the production of proinflammatory cytokines, which is related to the elevation of MAPK phosphorylation	[79]
Female BALB/c mice	Nanosilica	5, 10, and 20 mg/kg nanosilica through intranasal instillation	Exposure to nanosilica significantly elevated the characteristic markers of asthma including aryl hydrocarbon receptor (AHR), levels of inflammatory mediators and IgE, inflammatory cell infiltration, and mucus production	[80]
Female BALB/c mice	S-SNPs *; M-SNPs **; P-SNPs ***	Intranasally inoculated with nanosilica	Acute nanosilica exposure induced significant airway inflammation and further aggravated airway inflammation	[82]
Wistar rats	Amorphous silica nanoparticles	2, 5, and 10 mg/kg nanosilica through intratracheal instillation	Nanosilica exposure significantly decreased the levels of superoxide dismutase, glutathione peroxidase, and NO production, while increasing the production of malondialdehyde (MDA)	[85]
*ApoE*^−/−^ mice	Amorphous silica nanoparticles	1.5, 3.0, and 6.0 mg/kg nanosilica through intratracheal instillation once every 7 days and 12 times in total	The serum levels of total triglycerides and low-density lipoprotein cholesterol were elevated after nanosilica exposure	[88]
Male Tuck-Ordinary mice	Amorphous silica nanoparticles	Intraperitoneally administered with nanosilica at a dose of 0.25 mg/kg	Acute systemic exposure to nanosilica caused oxidative stress, inflammation, and DNA damage in multiple major organs, including the lungs, heart, liver, and brain	[93]
Male C57BI/6 mice	M-SNPs	50, 100, and 200 mg/kg of nanosilica through intragastric administration	Nanosilica affected the expression of metabolites involved in a series of metabolic pathways, including pyrimidine metabolism, purine metabolism, central carbon metabolism in cancer, protein digestion and absorption, and mineral absorption	[97]
Wistar rats	Amorphous silica nanoparticles	4 mg/dose nanosilica oropharyngeal aspiration twice a week	Nanosilica exposure caused kidney damage, with early tubular injury and inflammation	[98]

* Spherical silica nanoparticles, ** mesoporous silica nanoparticles, *** PEGylated silica nanoparticles.

**Table 5 ijms-25-07654-t005:** In vitro studies of cytotoxicity induced by PM2.5, silica, and nanosilica.

Cell Models	Types of Dust and Treatment	Main Results	Reference
Macrophages	Treated with 100, 200, and 400 μg/mL PM2.5 for 12, 24, and 48 h	The survival rate of macrophages decreased with increasing PM2.5 exposure time and dose; the levels of TNF-α and CRP in macrophages elevated	[109]
Beas-2b cells	Treated with 50, 100, 200, and 400 μg/mL PM2.5 for 24 h	PM2.5 significantly increased the levels of ROS in a dose-dependent manner, leading to NOX4/Nrf2 redox imbalance	[110]
A549 cells	Treated with 25, 50, 100, 200, and 400 μg/mL for PM2.5 24 h	PM2.5 activated the Keap1/Nrf2 pathway, leading to excessive intracellular ROS production and significant cytotoxicity in a dose-dependent manner	[111]
HBE cells	Treated with 16, 32, 64, and 128 μg/mL PM2.5 for 24 h	PM2.5 evidently induced viability inhibition, DNA damage, and part of apoptosis compared with the normal control in a forward concentration-dependent manner	[112]
N2a cells	Treated with 25, 50, 100, 200, and 400 μg/mL PM2.5 for 24 h	PM2.5 exposure induced a significant increase in intracellular ROS and disruption of physiological functions of macromolecules in a concentration-dependent manner	[113]
HaCaT cells	Treated with 5, 10, 25, 50, 100, 200, 300, 400, 500, and 800 μg/mL for 24 h	PM2.5 stimulated HaCaT cells to release a series of cytokines and proinflammatory cytokines, further stimulating epithelial cells, fibroblasts, and endothelial cells to secrete cytokines and adhesion molecules	[114]
HK-2 cells	Treated with 50 μg/mL PM2.5 for 0, 6, 12, and 24 h	The concentrations of IL-1β, TNF-α, and IL-6 in culture supernatants were significantly increased in HK-2 cells following PM2.5 exposure time-dependently	[115]
HEK-293 cells	Treated with 50, 100, and 200 μg/mL PM2.5 for 24 h.	The levels of IL-6, IL-18, TNF-α, and IL-1β were increased significantly relative to controls	[116]
HTR-8 cells	Treated with 1000, 5000, 10,000 ng/mL PM2.5 for 48 h.	The levels of IL-6 increased and the levels of hCG in cells decreased dose-dependently, resulting in a decrease in cellular viability and ER stress	[117]
EBf-H9 cells	Treated with 3.91, 7.81, 15.63, 31.25, 62.50, and 125.00 μg/cm^2^ of PM2.5 for 6 h	PM2.5 exposure reduced cell viability, increased lipid peroxidation levels, and induced ROS production, causing oxidative stress and cellular oxidative damage	[118]
HBE cells and MRC-5 cells	HBE cells were treated with 50 μg/mL silica for 24 h; the obtained supernatant was used to treat MRC-5 cells for 48 h	Silica increased the levels of collagen I (COL I) and α-smooth muscle actin (α-SMA) in MRC-5 cells and activated ferroptosis in HBE cells	[122]
A549, HBE and THP-1 cells	HBE and A549 cells were treated with 30, 50, 10, 150, and 200 μg/mL silica for 24 h or treated with 200 μg/mL silica for 12, 24, and 48 h separately; THP-1 cells were treated with 200 μg/mL silica for 12 h	Silica increased the expression of α-SMA and waveform protein and decreased the expression of E-calmodulin dose-dependently	[123]
Beas-2b cells and PBEC	BEAS-2B were exposed to 150 × 10^6^ μm^2^/cm^2^ and PBECs were exposed to 100 × 10^6^ μm^2^/cm^2^ crystalline silica	Crystalline silica induced canonical Wnt signaling (β-catenin) and decreased non-canonical (Wnt5A) signaling, contributing to proliferative, fibrogenic, and inflammatory responses in lung epithelial cells	[125]
BEAS-2B cells	Exposed to 25 μg/mL 41, 61, and 200 nm amorphous silica nanoparticles for 24 h	The genes involved in the immune and inflammatory response, gene expression, signal transduction, ER stress, oxidative stress, cell metabolism, and cell proliferation were gradually upregulated with the particle size decreasing	[128]
A549 cells	Exposed to 10, 50, and 100 μg/mL 15 nm and 46 nm amorphous silica nanoparticles for 12, 24, and 48h	Exposure to 15 nm nanosilica generated oxidative stress in A549 cells as reflected by reduced GSH levels, elevated production of MDA, and lactate dehydrogenase (LDH) leakage	[130]
HUVECs	Exposed to 50 μg/mL 60 nm nanosilica for 24 h	Nanosilica induced LDH release and oxidative injury, promoted the activation of Nrf2 signaling, promoted endothelial cell apoptosis and autophagy, and induced MMP collapse	[131]
HUVECs	Exposed to 12.5, 25, 50, and 100 μg/mL 58 nm Stöber silica nanoparticles for 24 h	Nanosilica induced ROS generation and caused redox imbalance, oxidative stress, and NO/NOS system disorder, leading to oxidative damage and inflammation response, further resulting in endothelial cytotoxicity and endothelial dysfunction via the MAPK-Nrf2 and NF-κB signaling pathways	[89]
HUVECs	Exposed to 25, 50, 100, and 200 μg/mL 20 and 100 nm S-SNPs * for 24 h	Nanosilica-100 at 100 μg/mL and 200 μg/mL was more toxic in inducing LDH release, decreasing the viability of HUVECs, and damaging the membrane integrity than nanosilica-20 at the same concentrations	[133]
N9 cells	Exposed to 50, 100, 150, and 200 μg/mL 50, 100, and 300 nm Stöber silica nanoparticles for 24 h	Nanosilica exhibited particle-size-dependent toxicity, leading to mitochondrial ROS production, expression of proinflammatory cytokines, and gaseous protein-d cleavage and pyroptosis	[134]
SH-SY5Y cells	Exposed to 3.125, 6.25, 12.5, 25, and 50 μg/mL 70 nm Stöber silica nanoparticles for 24 h	Nanosilica led to MMP loss, mitochondria damage, decreased cell viability, caused neural cell membrane damage, increased intracellular Ca^2+^ level, and elevated the expression of Apaf1 and Cyt C, which triggered caspase cascades of caspase-9 and caspase-3, ultimately resulting in neural apoptosis	[135]
HepG2	Exposed to 10, 25, 50, 100, and 200 μg/mL 15 nm amorphous silica nanoparticles for 24 h	Nanosilica exposure caused alterations in global metabolomics, specifically, the depletion of glutathione, NADPH oxidase-mediated ROS formation, and alteration in the antioxidant enzyme system	[136]
HepG2	Exposed to 1, 5, 10, 25, 50, 100, and 200 μg/mL 14nm amorphous silica nanoparticles for 72 h	Nanosilica-induced apoptosis in HepG2 may be mediated and regulated through the p53, Bax/Bcl-2, and cysteine asparaginase pathways	[137]
HL-7702 and BRL-3A cells	Exposed to 31.25, 62.5, 125, 250, and 500 μg/mL 20 nm amphipathic silica nanoparticles for 72 h	Nanosilica induced cytotoxicity and oxidative stress in HL-7702 and BRL-3A cells in dose-dependent manners through oxidative stress-mediated and p53, Bax/Bcl-2, and caspase-dependent pathways	[138]
Caco-2 and HT29-MTX co-cultures	Exposed to 2.65 g/cm^3^ 30 nm amorphous silica nanoparticles for 4 h or 5 d	Nanosilica significantly affected nutrient absorption and barrier function, decreased the number of intestinal microvilli, and significantly increased the brush border membrane enzyme intestinal alkaline phosphatase, further changed the expression levels of nutrient transport proteins, and induced ROS and proinflammatory signaling	[139]

* Spherical silica nanoparticles.

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
