# Peer review of "Impact of Respiratory Dust on Health: A Comparison Based on the Toxicity of PM2.5, Silica, and Nanosilica"

_ijms, 2024, doi:10.3390/ijms25147654_

Round 1

Reviewer 1 Report

Comments and Suggestions for Authors

In this review the authors  provide  a valuable update on toxicity of respiratory dust of different particle sizes, as PM2.5 (<2.5 μm), silica (<5 μm), and nanosilica (<100 nm) which are present in the enviroment. and derive   from various sources. Silica is considered a common pollutant derived from modern  industrial technology and so there is an increased warning for risk of occupational exposure. Indeed for its size (diameter of less than 5 microns) silica can penetrate deep into the lung bronchioles   and alveoli leading to severe diseases such as silicosis. PM2.5, which refers to particles with aerodynamic diameters less than 2.5 um, is believed to be one of the primary causes of respiratory and cardiovascular diseases. The rapid development of nanotechnology has resulted in the widespread use of nanosilica as a novel material in various fields, such as medicine, cosmetics, food, and agriculture. Moreover, nanosilica-induced toxicity can have devastating effects on the entire human system, such as epigenetic and phenotypic-related changes and immune system damage. Nanosilica presents small size and so it is thought to cause different negative consequences compared to silica. In order to comprehend the combined effects of simultaneous exposure to PM2.5, silica, and nanosilica in mixed environments and their respective mechanisms, the authors highlight that there is a need for more in-depth studies. To this end they evaluated the health consequences of respiratory dust on the human body and proposed potential synergistic effects based on data from in vivo and in vitro  studies.

The authors reported several  data on the environmental standards for PM2.5 derived from European Commission, United States Environmental Protection Agency, Ministry of the Environment Government of Japan, Ministry of Environment of South Korea, and Ministry of Ecology and Environment of the People’s Republic of China. The differences in values are a result of the distinct industrial conditions in one territory compared to another.    FDA has approved Nanosilica as an inorganic carrier in nanomedicine and this use can  increased the risk for human health since nanosilica is different from its natural form in terms of its surface-to-volume ratio, surface reactivity, and dispersing ability. Indeed the high potential toxicity could be explained by these properties.

  The authors reported many data on in vivo models, from rat to , zebrafish, pig, horse, camel and nonhuman-primate. The authors highlight that there are  inherent limitations of animal physiology, and all these data can not be representative for human exposure scenarios.

The in vitro models for assessing toxicity of PM25 are based on human-based cell cultures. Several evidences showed that PM25 can enter from respiratory tract to the blood circulatory   system, and affect the metabolism of all human systems, by triggering oxidative stress. The lung, kidney, bowel, cardiovascular systems have suffered significant damage. Moreover nanosilica can cross the placental barrier and reach the fetal bloodstream, leading to negative effects on embryonic development. Indeed also Blood Brain barrier can be crossed and so can lead to neurodegenerative diseases. 

taken togeteher the reported data, the authors summarize the several pathways that are affected by  PM2.5, silica and nanosilica: oxidative stress,   mitochondrial impairment, apoptosis, necrosis, DNA damage, mutagenesis, cancerogenesis, inflammation, release of proinflammatory cyokines, 

The authors continue to stress the need for conducting human-based studies on toxic effects of dust particles of different size

 The risk to human health from exposure not only to silica but above all to PM25 and nanosilica considering the recent use of the latter in several fields including nanotecnologies and nanomedicine. ThIIn this review, the authors reported many In vitro human-based studies on cytotoxicity induced by PM2.5, silica, and nanosilica as well in vivo evidences. The analysisi of the reported results  it possible to identify several molecular pathways that are involved in triggering the multiorgan toxic effects during exposure to such compounds.   The Authors should consider and suggest the potential for investigating these effects using the innovative models of reconstructed 3D human tissues such respiratory tract (bronchial, alveolar models), intestinal models, blood brain barrier and so on. These models are continuously developing since they are becoming more and more important because they enable us to validate the effects of chemical compounds and arrange them in a sequence to simulate the cross-talk between cells and tissues.the In my opinion, the review shows a relevance since it  reports a careful report of the literature on toxicity of dust small size elements, such PM25 and nanosilica. Highlighting the capability of small size silica to cross anatomical barriers, such our intestinal wall and this intestinal barrier and as a result  can find itself in the blood circulation reaching all organs. Overcoming of the blood-brain barrier, facilitated by the inflammatory process triggered at the intestinal and endotelial level overcoming the blood-brain barrier may trigger those neurodegenerative processes that seem to be related as consequence of such  exposure.   Taken together the evidence and arguments presented, the authors presented consistent conclusions, but have to stressed the need to deep the risks of exposure to these silica-derived compounds by using 3D human-based models as well organoids to better identified the molecular mechanisms which underline the effects on tissues, organs as well as the adverse outcoming effects. Considering the concept of OneHealth it is crucial to examine the effects not only on humans but also on the environment. Although not extensively discussed in the review, this aspect is important due to the risks to the environment, plants, and animals that could have additional repercussions for humans.   The table on Environmental standards for PM2.5 in different regions shows utmost care, providing information on the most important aspects of exposure limits.  The tables on in vivo and in vitro data In vitro and in vivo toxicity data tables provide many important details that shed light on the results. References are appropriate       I think that this review  opens up new challenges for more and more in-depth research in order to to answer the issues, tobetter predict and support exposure risks,  by using innovative approaches in vitro as well in silico.

Author Response

Thank you for your letter and reviewer’s comments concerning our manuscript entitled “Impact of Respiratory Dust on Health: A Comparison Based on the Toxicity of PM2.5, Silica, and Nanosilica” (ID: ijms-3053323). These comments are all valuable and very helpful for revising and improving our paper, as well as the important guiding significance to our review.

Reviewer 2 Report

Comments and Suggestions for Authors

It is a very interesting paper about the impact of silica/PM2.5/nanosilica, thus respiratory dust on human health. It is easy to follow and provides the newest research findings on the subject. There are only few issues I would like to address to the authors.

Line 41 - should it be “human system” or “human body”?

Line 69 – what do you understand here under the term “metals”? Please explain and provide a better term if possible.

Correct and unify all sub/superscripts in the entire text.

Line 254 – what is the exact link between silica and silicosis?

Line 262-3 – write this part more clearly. Is silicic acid really produced from dissolved silica? Under which conditions? Please check and explain a bit.

Explain all abbreviations of cell lines.

Explain the term “IARC”.

Line 661 – explain/write in a clear way the relationship between “Desert Storm" (military operation during the Gulf War) and silica/PM2.5 exposure.

Consider to add to the conclusions a small paragraph about methods of protection from PM2.5/silica/nanosilica.

Author Response

Thank you for your letter and reviewer’s comments concerning our manuscript entitled “Impact of Respiratory Dust on Health: A Comparison Based on the Toxicity of PM2.5, Silica, and Nanosilica” (ID: ijms-3053323). These comments are all valuable and very helpful for revising and improving our paper, as well as the important guiding significance to our review. We have studied comments carefully and have made a correction which we hope meet with approval. Revised portions are marked in yellow on the paper. The main corrections in the paper and the responses to the reviewer’s comments are as follows:

Point 1. Line 41 - should it be “human system” or “human body”?
Response 1:

  Thanks for your question. Since nanosilica can enter the circulation through the respiratory tract and have adverse effects on multiple systems of the human body, we think it is more accurate to use "human system" in this sentence.

Point 2. Line 69 – what do you understand here under the term “metals”? Please explain and provide a better term if possible.
Response 2:

  Thanks for your question. “Metals” here refer to metals that can adhere to PM2.5, including Pb, Ni, Ca, Cu, Fe, Si, and so on. Due to the complex composition, including heavy metals, light metals and metalloids, we think it is more accurate to use “metals” here.

Point 3. Correct and unify all sub/superscripts in the entire text.
Response 3:

  We’re very sorry for this oversight. We have made modifications and marked them in yellow.

Revised: Currently, many regions have established 1-hour average concentration limits for gaseous air pollutants such as SO2, NO2, CO and O3.

µg/m³, mg/m³, g/m³.

Point 4. Line 254 – what is the exact link between silica and silicosis?
Response 4:

  We are very sorry for this unclear expression which may have caused confusion. Silicosis is a diffuse interstitial pulmonary fibrosis caused by prolonged inhalation of silica. We have modified the sentences and marked them in yellow.

Revised: The lung, as the target of silica action in the body, is under the greatest threat of toxicity and is susceptible to injury. Prolonged inhalation of free silica can lead to silicosis [62].

Point 5. Line 262-3 – write this part more clearly. Is silicic acid really produced from dissolved silica? Under which conditions? Please check and explain a bit.
Response 5:

  Thanks for your question. Silicic acid is produced by the dissolution of silicon dioxide in water, that is, by a hydration reaction. As the specific process is more complicated, it needs a lot of words. The focus of our review is not on silicic acid, if you are interested, you can refer to the descriptions in these three articles.

[1] Carlisle, E.M. Silicon. Nutr. Rev. 1975, 33, 257–261.

[2] Wang, L.Y.; Scabilloni, J.F.; Antonini, J.M.; Rojanasakul, Y.; Castranova, V.; Mercer, R.R. Induction of secondary apoptosis, inflammation, and lung fibrosis after intratracheal instillation of apoptotic cells in rats. Am. J. Physiol. Lung C 2006, 290, L695–L702, doi:10.1152/ajplung.00245.2005.

[3] Hamilton, R.F., Jr.; Thakur, S.A.; Holian, A. Silica binding and toxicity in alveolar macrophages. Free Radic. Biol. Med. 2008, 44, 1246–1258, doi:10.1016/j.freeradbiomed.2007.12.027.

Point 6 Explain all abbreviations of cell lines.
Response 6:

  We’re very sorry for the mistake. We have explained all abbreviations of cell lines, adjusted their formatting in the review uniformly, added them to the Alphabetical list of abbreviations, and marked it in yellow.

Revised:

Alphabetical list of abbreviations

16HBE cells

Normal human bronchial epithelial cells

A549 cells

Human lung adenocarcinoma epithelial cells

AHR

Aryl hydrocarbon receptor

AM

Alveolar macrophage

AMP

Adenosine 5‘-monophosphate

AMPK

AMP-activated protein kinase

ANS

Autonomic nervous system

Apaf1

Apoptotic protease activating factor-1

ARE

Antioxidant response element

BALF

Bronchoalveolar lavage fluid

BAX

Bcl-2-associated X protein

Bcl-2

B-cell lymphoma 2

Beas-2b cells

Human normal lung epithelial cells

BRL-3A cells

Rat normal liver cells

Caco-2 cells

Human intestinal epithelial cells

CD36

Platelet glycoprotein 4

CNS

Central nervous system

COL I

Collagen I

CVD

Cardiovascular disease

Cyt C

Cytochrome C

EBf-H9 cells

Human embryonic stem cells

ECM

Extracellular matrix

EMT

Epithelial-mesenchymal transition

ER

Endoplasmic reticulum

ERK1

Extracellular signal-regulated kinase 1

FDA

Food and Drug Administration

GSDMD

Gasdermin D

GSH

Glutathione

HaCaT cells

Human keratinocyte-forming cells

HEK-293 cells

Human embryonic renal cells

HepG2 cells

Human liver cancer cells

HK-2 cells

Human renal cortical proximal tubule epithelial cells

HL-7702 cells

Human normal liver cells

HRV

Heart rate variability

Hs-CRP

High-sensitivity C-reactive protein

HT-29 cells

Human intestinal epithelial cells

HTR-8/SVneo cells

Human chorionic trophoblast cells

HUVECs

Human umbilical vein endothelial cells

IARC

International Agency for Research on Cancer

IER

integrated exposure-response

IL-1β

Interleukin-1β

IL-6

Interleukin-6

iNOS

Inducible nitric oxide synthase

Keap1

Kelch-like ECH-associated protein 1

LDH

Lactate dehydrogenase

MAPK

Mitogen-activated protein kinase

MDA

Malondialdehyde.

MMP

Mitochondrial membrane potential

MRC-5 cells

Human lung fibroblasts

M-SNPs

Mesoporous silica nanoparticles

mtROS

Mitochondrial reactive oxygen species

MTX

Methothexate

N2a cells

Mouse neuroblastoma cells

N9 cells

Microglia cells

NF-κB

Nuclear factor kappa-B

NO

Nitric oxide

NOX4

NADPH oxidase

Nrf2

NF-E2 p45-related factor 2

OELs

Occupational exposure limits

OSHA

Occupational Safety and Health Administration

p62

Sequestosome 1

PAHs

Polycyclic aromatic hydrocarbons

PBEC

Primary human bronchial epithelial cells

PC-TWA

Permissible concentration time-weighted average

P-SNPs

PEGylated silica nanoparticles

RAW264.7

Mouse monocyte macrophages

ROS

Reactive oxygen species

SH-SY5Y

Human neuroblastoma cells

SPM

Suspended particulate matter

S-SNPs

Spherical silica nanoparticles

TGF-β1

Transforming growth factor 1

THP-1 cells

Human leukemia monocytic cells

TNF-α

Tumor necrosis factor α

VCAM-1

Vascular cell adhesion molecule-l

WHO

World Health Organization

WNT

Wingless-type MMTV-integration site

α-SMA

α-smooth muscle actin

Point 7 Explain the term “IARC”.
Response 7:

We’re very sorry for this oversight. We have made modifications and marked them in yellow.

Revised:

Silica has been classified as a Group 1 carcinogen by International Agency for Research on Cancer (IARC) since 1997.

Point 8 Line 661 – explain/write in a clear way the relationship between “Desert Storm" (military operation during the Gulf War) and silica/PM2.5 exposure.
Response 8:

Thanks for your sincere suggestion. Since “Desert Storm” contains both silica and PM2.5, we thought it could be used to discuss combined exposure of silica and PM2.5.

Point 9 Consider to add to the conclusions a small paragraph about methods of protection from PM2.5/silica/nanosilica.
Response 9: Thanks for your sincere suggestion. We have made corresponding modifications according to your suggestion and marked them in yellow.

Revised:

Besides enhancing toxicity studies, we suggest implementing more policies aimed at decreasing environmental levels of PM2.5, silica, and nanosilica, specifically by revising exposure limits to minimize overall population exposure.
